# Automatic Segmentation with Deep Learning in Radiotherapy

**DOI:** 10.3390/cancers15174389

**Published:** 2023-09-01

**Authors:** Lars Johannes Isaksson, Paul Summers, Federico Mastroleo, Giulia Marvaso, Giulia Corrao, Maria Giulia Vincini, Mattia Zaffaroni, Francesco Ceci, Giuseppe Petralia, Roberto Orecchia, Barbara Alicja Jereczek-Fossa

**Affiliations:** 1Division of Radiation Oncology, IEO European Institute of Oncology IRCCS, 20141 Milan, Italy; larsjohannes.isaksson@ieo.it (L.J.I.); federico.mastroleo@ieo.it (F.M.); giulia.corrao@ieo.it (G.C.); mariagiulia.vincini@ieo.it (M.G.V.); mattia.zaffaroni@ieo.it (M.Z.); barbara.jereczek@ieo.it (B.A.J.-F.); 2Department of Oncology and Hemato-Oncology, University of Milan, 20141 Milan, Italy; francesco.ceci@ieo.it (F.C.); giuseppe.petralia@ieo.it (G.P.); 3Division of Radiology, IEO European Institute of Oncology IRCCS, 20141 Milan, Italy; paul.summers@ieo.it; 4Department of Translational Medicine, University of Piemonte Orientale (UPO), 20188 Novara, Italy; 5Division of Nuclear Medicine, IEO European Institute of Oncology IRCCS, 20141 Milan, Italy; 6Precision Imaging and Research Unit, Department of Medical Imaging and Radiation Sciences, IEO European Institute of Oncology IRCCS, 20141 Milan, Italy; 7Scientific Directorate, IEO European Institute of Oncology IRCCS, 20141 Milan, Italy; roberto.orecchia@ieo.it

**Keywords:** radiotherapy, segmentation, automatic, deep learning, artificial intelligence, artificial neural networks

## Abstract

**Simple Summary:**

Automatic segmentation of organs and other regions of interest is a promising approach for reducing the workload of doctors in radiotherapeutic planning, but it can be hard for doctors and researchers to keep up with current developments. This review evaluates 807 papers and reveals trends, commonalities, and gaps in the existing corpus. A set of recommendations for conducting effective segmentation studies is also provided.

**Abstract:**

This review provides a formal overview of current automatic segmentation studies that use deep learning in radiotherapy. It covers 807 published papers and includes multiple cancer sites, image types (CT/MRI/PET), and segmentation methods. We collect key statistics about the papers to uncover commonalities, trends, and methods, and identify areas where more research might be needed. Moreover, we analyzed the corpus by posing explicit questions aimed at providing high-quality and actionable insights, including: “What should researchers think about when starting a segmentation study?”, “How can research practices in medical image segmentation be improved?”, “What is missing from the current corpus?”, and more. This allowed us to provide practical guidelines on how to conduct a good segmentation study in today’s competitive environment that will be useful for future research within the field, regardless of the specific radiotherapeutic subfield. To aid in our analysis, we used the large language model ChatGPT to condense information.

## 1. Introduction

Radiotherapy (RT) represents an important treatment modality for cancer patients that uses high-dose ionizing radiation for curative or palliative purposes. When RT is prescribed, accurate delineation of the tumors, target volumes, and organs at risk (OARs) is crucial for effective treatment planning and reducing the risk of radiation-induced toxicity. However, manual segmentation of these structures, which is the current standard of care, is time-consuming and prone to inter- and intra-observer variability [1,2,3,4,5,6,7], which can contribute to suboptimal treatment outcomes. Moreover, the task is repetitive and tiresome, which makes the process tedious for doctors, who typically prefer to devote their time elsewhere. Thus, there is a significant need for good automatic segmentation methods that can be incorporated into clinical practice to improve current workflows and outcomes.

The repetitive nature of the segmentation process makes it a prime candidate for automation with modern machine learning methods, which excel in performing monotonous information-based tasks quickly. Recent years have seen deep learning (DL)-based autosegmentation methods gain significant attention in the field of radiotherapy (see results below). These methods typically employ convolutional neural networks (CNNs) to automatically learn relevant features from medical images which helps them segment the structures accurately. The increasing interest in DL-based automatic segmentation is driven by its success over traditional methods such as atlases, active contours, or region-growing, particularly as demonstrated in online segmentation challenges like BraTS [8], KiTS [9], AMOS [10], and PROMISE [11]. The advantages of DL-based methods include improved accuracy, reproducibility, speed, observer variability, and efficiency, which are all crucial factors in radiotherapy treatment planning. But despite the many advantages, several limitations exist, most notably the requirement for large, annotated training datasets, which can be time-consuming and resource-intensive to collect. Another limitation is the potential for overfitting and related generalization issues, which can lead to erroneous segmentation results when the model encounters atypical samples [12,13]. Some notable developments such as transfer learning [14], data augmentation [15], and ensemble models [16] may aid in overcoming these limitations. Transfer learning has shown promise in reducing the amount of required training data, while data augmentation techniques can increase the diversity of the training dataset at hand. Ensemble models can be used to improve both segmentation accuracy and robustness by combining multiple models into a better performing ensemble (e.g., using models with different architectures or training parameters).

The problem setting for automatic segmentation varies greatly, partly because RT is applied to treat various types of cancer located in many different body regions, including the brain, breast, prostate, lung, and others. Multiple different image types, including magnetic resonance imaging (MRI), computed tomography (CT), and positron emission tomography (PET), are all frequently utilized to visualize the tumor and surrounding normal tissues in the treatment planning process. Each imaging modality excels at depicting different information, which means that they are used to varying extents in different cancer sites. However, most DL models are modality agnostic, meaning that they can be adequately applied to different image types provided that they have been trained on similar images. Therefore, much can be learned from observing advances in adjacent fields, even for researchers working exclusively on a single cancer site or image modality. Another important source of variability relies on the different pathological conditions and physical characteristics of the patients. Automatic segmentation is further complicated by the fact that imaging protocols may vary between institutions or even according to who performs the examination. Therefore, developing robust and versatile automatic segmentation methods that can handle different cancer sites and imaging modalities is essential for their success in radiotherapy treatment planning and execution.

The aim of this review is to analyze the current state of the field of medical image segmentation with deep learning for RT and attempt to answer questions about general trends and needs. We are particularly interested in the datasets, images, cancer sites, and limitations. Our intention is to help identify areas where more research might be needed, to illuminate common flaws, and to guide new research in the field. To do this, we pose explicit questions aimed at providing concrete overviews of the common trends and methods. The analysis we put forward helped us distill a set of four key points that we believe can be useful in designing new studies and maximizing the impact of future work in the field.

## 2. Related Work

Many reviews have been published on automatic segmentation methods for specific regions of interest [17,18,19,20,21,22,23,24,25,26], image modalities [17,20,24,27,28], and methods [29,30,31,32,33,34,35,36,37,38,39,40]. However, none of these thoroughly cover all three aspects to provide a holistic overview of the state of the field. Moreover, previous reviews tend to focus on quantitative analysis of a relatively limited set of papers, or the different methods and models applied within them. 

Our contribution to this work is three-fold. First, we gathered a very large set of 807 papers and collected statistics to provide a comprehensive overview of the field of medical image segmentation in radiotherapy. Second, we highlight a novel way to leverage modern large language models (LLMs) like ChatGPT for condensing large bodies of information. Third, we pose explicit questions for the analysis to provide a valuable resource for researchers seeking actionable insights.

## 3. Data Collection

To collect relevant academic papers in the field of automatic segmentation with deep learning in radiotherapy, we searched the Scopus and PubMed databases using the following query and collected all unique results:Title: “segmenting” or “*segmentation”;Title, abstract, or keywords: “CT”, “MRI”, “PET”, “DWI”, or “*medical image*”;Title, abstract, or keywords: “deep learning” or “artificial neural network*”;Title, abstract, or keywords: “radiotherapy”;Document type: article or conference proceeding;Language: English.

All review articles and unrelated papers were removed from the dataset. The relevant metadata (including abstract, publication date, URL, and citation information) were collected for all remaining publications using both web-scraping tools and manual annotations.

As a data mining tool, we utilized the large language model ChatGPT (v3.5-turbo, Mar 14 version, OpenAI [41]). This model is designed to respond to arbitrary natural language queries by producing text outputs. The model was asked to answer the following questions about the abstract of each paper (the answers can be found online in the Appendix A):Is the paper a review article? (Yes/No).How many patients were used in the study? Only answer with a number.What type of images were used in the study? Choose from MRI/PET/CT/DWI/Ultrasound/Other.Did the paper use images from multiple sources or multiple organs? (Yes/No).Is the paper about organ or tumor segmentation? Choose from Organ/Tumor/Both.Did the paper propose a novel segmentation method or deep learning architecture? (Yes/No).Was the code made public? (Yes/No).Was the data made public? (Yes/No).What is the key takeaway of the paper? Answer in one sentence.

The responses were then reviewed such that the answers were correct and consistent. To instruct the language model to answer properly, we gave it the following system prompt (this prompt acts as a form of base instruction that the model takes into account when answering the users’ queries):

“*You will be asked to read a scientific paper and then answer questions about it. The paper is about automatic segmentation with deep learning in radiotherapy. Please prioritize correctness in your answers. If you don’t know the answer, respond with ‘I don’t know.’*”

## 4. Analysis

### 4.1. Statistics

Statistics about the publication year, number of included patients, target site, imaging modality (MRI/CT/PET/Ultrasound), and segmentation type (organ/tumor) were collected and plotted. To gauge the extent of research conducted on various topics of interest, we scanned the abstracts and GPT responses for the appearance of certain keywords relating to each topic. These topics included: transfer learning or pretraining, self-supervised or semi-supervised or contrastive learning, cross-validation, transformers, publicly available data, and open-sourcing code.

### 4.2. Region-Specific Analysis

All papers related to each body region (head and neck, brain, prostate, lung, cervix and uterus, liver, heart, stomach and colon, kidney, breast, pancreas, spleen, and skin) were also isolated and analyzed separately to uncover what research patterns and trends are common within the different cancer sites. This includes the images used, the organs and structures segmented, the methods employed, and the most common results and conclusions mentioned in the abstracts.

### 4.3. Subjective Analysis

To support and encourage further research in the field, we attempted to distill more actionable material from the collected papers by posing and discussing explicit questions aimed at providing more guidance. The questions were:What is missing from the current corpus?What should researchers think about when starting a segmentation study?How can research practices in medical image segmentation be improved?Do authors agree on conclusions, and is it possible to spot trends in the employed methods?

## 5. Results

### 5.1. Statistics

A flowchart of the paper and data collection process is shown in Figure 1. A total of 905 unique papers matched the query. After removing review articles and unrelated papers (e.g., papers in foreign languages or papers we were unable to access), 807 remained. The number of studies published per year is plotted in Figure 2. The field has been growing steadily since its mainstream adoption around 2013 (not counting five early adopters in 1999 [42], 2006 [43], 2008 [44], and 2010 [45,46]), accumulating a total of 225 publications in 2022 alone. The five most common cancer sites were: head and neck (H and N; 198 papers), brain (130 papers), prostate (117 papers), lung (107 papers), and cervix (101 papers; see Figure 3). The H and N area further stands out as having the most structures of interest for segmentation (e.g., larynx, pharynx, parotid glands, esophagus, masseter, thorax, etc.). Less frequently studied cancer sites included: the skin (6 papers), spleen (12 papers), pancreas (30 papers), breast (30 papers), and kidney (32 papers). In Figure 4, the number of publications per imaging modality and segmentation type is plotted. Roughly 58% of the publications (457 papers) studied CT images, 33% studied MRI (254 papers, of which 22 were specific to DWI), 8% studied PET images (64 papers), and 22 (3%) studied ultrasound images. The majority of the studies focused on organ segmentation (415 papers, or 56%) rather than tumor segmentation (260 studies, or 35%), with only about 9% (68 studies) focusing on both. A spreadsheet of the included studies along with abstracts, the generated answers, one-sentence summaries, and corresponding metadata can be found in Appendix A.

The combined studies yielded a median and mean patient count of 128 and 363, respectively (in cases where a study used multiple datasets, we used the total sum of patients). A graph showing the distribution of sample sizes is shown in Figure 5. Only 407 papers disclosed the sample size in the abstract. Of these, roughly 40% had fewer than 100 patients, and 42% were in the 100–500 patients range. The three cancer regions with the largest cohorts, as measured by median sample size (see Figure 6), were the brain (250 patients), spleen (215 patients), and lung (144 patients). The sites with the smallest cohorts were the heart (66 patients), liver (72 patients), and H and N (87 patients). Notably, heart and H and N also had the studies with the second and third largest sample sizes (6535 [47] and 6000 [48] patients, respectively)—not too far off from the largest sample size of 9032 [49].

When asked whether the study proposes a novel deep learning model or architecture, the language model responded “Yes” for 550 studies (70%) and “No” for only 232 studies (see Figure 7), revealing a strong tendency to introduce new methods.

### 5.2. Region-Specific Analysis

#### 5.2.1. Head and Neck

Within the head and neck region, 48 studies specifically concerned segmentation of the tumors while 59 segmented the pharynx, 41 segmented the esophagus, 32 segmented the parotid gland, 8 segmented the larynx, and 3 segmented the masseter. The most common architecture by far was a standard 3D U-net, but a notable fraction also employed GAN architectures while a few used 2D networks. A total of 64 papers used data from multiple sources or organs. In all papers, deep learning segmentation was superior to the competing method. Even though the head and neck region is one of the most studied body regions, it is one of the regions suffering the most from small data sets (median sample size 87).

#### 5.2.2. Brain

In the brain, the vast majority (94 out of 130) of studies focused on tumor segmentation. Only a handful focused on specific structures like the brain stem, pituitary gland, cerebellum, etc. Papers published regarding brain segmentation appeared to propose novel segmentation methods to a larger extent compared to the other regions; only 39 papers do not make specific mention of a new contribution. The 3D U-net was again the most common architecture, but many alternative approaches exist. Only 31 papers tested their methods on multiple datasets or organs. Apart from 27 studies that were inconclusive, all papers concluded that deep learning was superior to the alternative approach.

#### 5.2.3. Prostate

For the prostate, only 19 out of 117 studies focused on tumor segmentation while the majority focused on segmentation of the whole prostate. Many studies also performed segmentation of surrounding organs like the bladder (68 studies) and rectum (57 studies). Although the majority of papers focused on 3D networks, 12 papers also tested 2D models. Only 30 papers tested their methods on multiple datasets or organs. All but 5 out of the 117 studies concluded that deep learning performed better than other methods.

#### 5.2.4. Lung

The papers studying lung segmentation were dominated by 3D U-net type architectures, with a handful of 2D hybrid networks. Out of the 107 papers, roughly 80 proposed new architectures and 49 focused on tumor segmentation. A total of 9 papers used MRI images, 14 used PET, and 89 used CT images. All but one lung segmentation study that compared deep learning with other approaches concluded that the performance of deep learning was superior.

#### 5.2.5. Liver

Almost all studies of the liver analyzed CT images, but 11 also analyzed MRI images, 5 analyzed PET images, and 1 analyzed ultrasound images. A total of 49 papers studied the segmentation of tumors (20 of which studied segmentation of both tumors and organs); almost 60% of the 83 papers. Only 19 papers did not propose a new method or architecture, but 36 used images from multiple sources or organs. All but four papers concluded that deep learning was better than the alternative approach.

#### 5.2.6. Heart

For the heart, 53 out of 61 papers studied CT images while 2, 4, and 1 paper studied PET, MRI, and ultrasound images, respectively. Only 12 papers studied tumor segmentation, meaning that most papers focused on segmenting the heart and its various substructures. About 42 papers proposed a novel method or architecture while 18 did not. Two papers were inconclusive as to whether deep learning performed better than the competing approach and fifty-eight studies concluded that deep learning was better.

#### 5.2.7. Colon

Among the papers studying the colon, 41 focused on CT images and 12 focused on MRI images. Only a total of 12 out of 54 studies focused on tumor segmentation. Roughly two-thirds (41) of the papers proposed novel methods or architectures and 26 used images from multiple sources or organs. Most studies appeared to use fairly standard 3D U-net-type architectures. In total, 52 papers concluded that deep learning was superior (two studies were inconclusive).

#### 5.2.8. Kidney

Among the kidney studies, 3 out of 32 papers studied tumor segmentation only, while 4 papers studied both rumor and organ segmentation. Four studies used MRI images and twenty-four studies used CT images. Sixteen papers tested their methods on multiple datasets or organs. All but seven studies proposed novel methods and architectures and all but one study (which was inconclusive) concluded that deep learning was superior to alternative approaches.

#### 5.2.9. Breast

Within the breast studies, 14 used CT images, 8 used MRI images, and 5 used ultrasound images. Eleven out of the thirty studies focused on tumor segmentation, while four focused on both tumor and organ segmentation. Nine papers tested their methods on multiple datasets or organs. Twenty-two (22) studies proposed novel methods or architectures and all but two inconclusive studies concluded that deep learning outperformed alternative methods.

#### 5.2.10. Pancreas

In studies of the pancreas, 21 papers used CT images and 8 used MRI. Seven out of the thirty papers focused on tumor segmentation and only five did not explicitly propose new methods or deep learning architectures. More than half (18 papers) used data from multiple organs. All studies (except for two that were inconclusive) showed that deep learning outperformed conventional approaches.

#### 5.2.11. Cervix

Among the 26 studies of the cervix and uterus, 15 used CT images, 1 used PET images, 7 used MRI images, and 1 used ultrasound. Thirteen papers focused on tumor segmentation, seven on organ segmentation, and six on both. A total of nine papers did not propose any new methods or architectures. All studies concluded that deep learning was superior to competing approaches. 

#### 5.2.12. Spleen

Of the twelve papers studying the spleen, eight used CT images, and two used MRI images. Eight papers specifically mention developing new deep learning architectures or methods. None of them were specifically testing methods for automatic tumor segmentation, but most (nine papers) used data from multiple organs to test their methods. In total, all papers concluded that deep learning was superior to the alternative, apart from one study that was inconclusive. 

#### 5.2.13. Skin

In studies of the skin, three papers used ultrasound images, one used MRI images, one used dermoscopic images, and one used photographs of the skin. All six papers explicitly examined the segmentations of tumors, while half of them proposed novel methods. Five studies concluded that deep learning was superior to the alternative approaches.

### 5.3. Subjective Analysis

#### 5.3.1. What Is Missing from the Current Corpus?

Benchmark datasets: At present, direct comparisons of results remain unfeasible since very few studies use the same data. Specifically, only 92 studies employed data sourced from at least one of the following datasets: UK Biobank, BraTS, KiTS, HECKTOR, OASIS, or PROMISE. Moreover, there is a very strong preference for novelty over replication in the current corpus. This prevents researchers from knowing how their model fares against other models without having to implement them themselves (often from scratch), which in itself is often not possible due to insufficient details having been given in the publications of other groups. Efforts to publish high-quality benchmarks, analogous to the CIFAR10 and ImageNet datasets in general computer vision, are much more likely to yield high-impact results, especially since the literature is dominated by presentations of novel “state of the art” models. It is possible that this overabundance is a direct consequence of the lack of benchmarks, possibly convincing researchers that minor improvements on very small datasets result in a state-of-the-art model (while, in reality, it may just be the result of statistical variations).Open sourcing and making code available: Out of 807 papers, only three made code of their models available; an abysmal number compared to numbers in related research disciplines. Open-sourcing code is a great way to promote research and enable other researchers to benchmark their models and implement strong baselines, ultimately driving the field forward. If there are privacy objections to open-sourcing, a good compromise is to publish the code without the model weights.More research on, e.g., spleen and pancreas: As seen clearly in Figure 3, relatively few papers have been published on the spleen and pancreas compared to, e.g., H and N or lung cancer. It might be worthwhile for institutions to put more effort into collecting data in these areas so models and methods can be tested in a wider variety of settings.Evaluation of alternative training techniques: Computer vision is a rapidly evolving field with successful novel paradigms being introduced relatively frequently. Some notable training strategies appear to be less studied in the field of medical image segmentation. In particular, only 48 papers made use of transfer learning or pretraining, only 21 papers studied self-supervised, semi-supervised, or contrastive learning, and only 11 papers explicitly studied transformer-based architectures. It is possible that segmenting medical images could be more amenable to these methods than what is currently being realized.

#### 5.3.2. What Should Researchers Consider When Starting a Segmentation Study?

Collect more (high-quality) data: An alarming number of studies use very small datasets: 163 studies used fewer than 100 patients. While this can be good as a proof of concept, it is hard to draw conclusions from such a narrow scope. Furthermore, increasing the number of data samples can often be more effective than developing a stronger training pipeline (e.g., a better model) when it comes to both performance and generalizability.Test models on multiple datasets and cancer sites: Diversifying a model by training it on qualitatively different images is a great way to demonstrate the capacity of a model, even if it is intended for a very specific purpose. Moreover, models trained on multiple modalities (e.g., both images and patient data) often show superior performance compared to single-modality training [50,51,52,53,54]. One way to do this is to include images from open sources like the UK Biobank, the Cancer Imaging Archive (TCIA), the BraTS (brain tumor segmentation) dataset, the MM-WHS (Multi-modality whole heart segmentation) dataset, and even The Cancer Genome Atlas (TCGA).Evaluate existing models and training techniques over developing new ones: There is a relative overabundance of papers proposing novel deep learning architectures/modules and claiming state-of-the-art performance on rather narrow datasets. It seems exceedingly unlikely that a model trained on just about 50 patients can really be considered state-of-the-art within a certain field. As such, the scientific community should focus more on evaluating existing models and training techniques on larger and more representative datasets and benchmarking them against established baselines.Focus on clinical viability over minor improvements in metrics: A limited number of publications assess the practical feasibility of their models, such as subjecting them to downstream applications, investigating their generalizability across different patient populations and clinical scenarios, or evaluating the contours they produce from a medical standpoint. It seems likely that the difficult legislative environment and distance to market limits the number of groups willing to dedicate time and energy to rendering their models suitable for clinics. The regulatory landscape for healthcare is complex and can be a significant obstacle to the implementation of new technologies, which is why research on these issues is important.

#### 5.3.3. How Can Research Practices in Medical Image Segmentation Be Improved?

In addition to the general remarks made under the previous two headlines, we suggest using the following good practices, which often appear to be overlooked, to improve the methodology in individual studies: Follow good reporting/documentation guidelines: We found a surprising number of studies lacking critical information needed to reproduce or even implement the methods given in the study. In particular, multiple studies (and abstracts) fail to report details about the patients (e.g., the class balance), the training procedure (e.g., learning rate, data augmentation parameters, and dropout rate), or the validation procedure (e.g., whether the train-test split was random or not). A good place to start is to follow proposed guidelines such as [55] or [56].Do cross-validation: Despite being a standard practice in machine learning, a surprising number of studies do not cross-validate their models and instead opt to use a single train-test split. This severely limits the validity of the conclusions, particularly for smaller datasets, and is strongly discouraged.Implement more robust baselines: Implementing a strong baseline is critical in order to adequately assess a new model (particularly due to the lack of open benchmarks). Yet, a number of studies appear to use dated or suboptimally implemented references such as a naïve U-net. A good place to start is models like SegFormer [57] or nn-Unet [58], which are open-source and can be copied directly from their respective repositories.

#### 5.3.4. Do Authors Agree on Conclusions, and Is It Possible to Spot Trends in the Employed Methods?

Authors overwhelmingly agree that deep learning has the potential to greatly help in the process of delineating regions of interest in medical images—not a single paper reviewed in this study specifically concluded that deep learning was actively inferior to traditional approaches. However, very few papers study well-established model architectures on diverse datasets and body regions. Thus, instead of seeing the field converge on very strong models with wide applicability—like the transformer network in natural language processing—we see a continuing trend to propose novel architectures that perform very well on a small, constrained dataset.

## 6. Limitations

This review contains both objective and subjective analyses and may be considered both systematic and narrative, or neither. To prevent a biased selection and interpretation, we employed an automated selection procedure without hand-picking papers to exclude or include. As a result, some papers from the search results may be less relevant than others. Moreover, due to the large number of papers and queries answered by ChatGPT, we did not check the correctness of all answers. In our quality control of about 660 answers, the model only made one error, where it conflated the sample size and the number of images in the study. Lastly, the model was only presented with the abstracts of the papers.

## 7. Conclusions

In this review, we have analyzed the current state of research in medical image segmentation by examining 807 studies that make use of deep learning for radiotherapy applications. Our statistical evaluation revealed that 58% of the publications were based on CT, 33% on MRI, and 8% on PET imaging. The three most studied cancer areas were the head and neck, lung, and brain regions, while the three least studied areas were the spleen, pancreas, and kidney. The median number of patients across all studies was 137. Several issues in the field were highlighted, including the lack of benchmark datasets, the need for open-sourcing code, the importance of collecting more high-quality data, and the need to evaluate existing models and training techniques instead of developing new ones. We also suggested good practices to improve methodology in individual studies, such as following good reporting/documentation guidelines, conducting cross-validation, and implementing more robust baselines. Overall, we hope that this provides valuable insights into the challenges and opportunities in medical image segmentation research and that our suggestions can help researchers improve the quality and rigor of their future work.

## Figures and Tables

**Figure 1 cancers-15-04389-f001:**
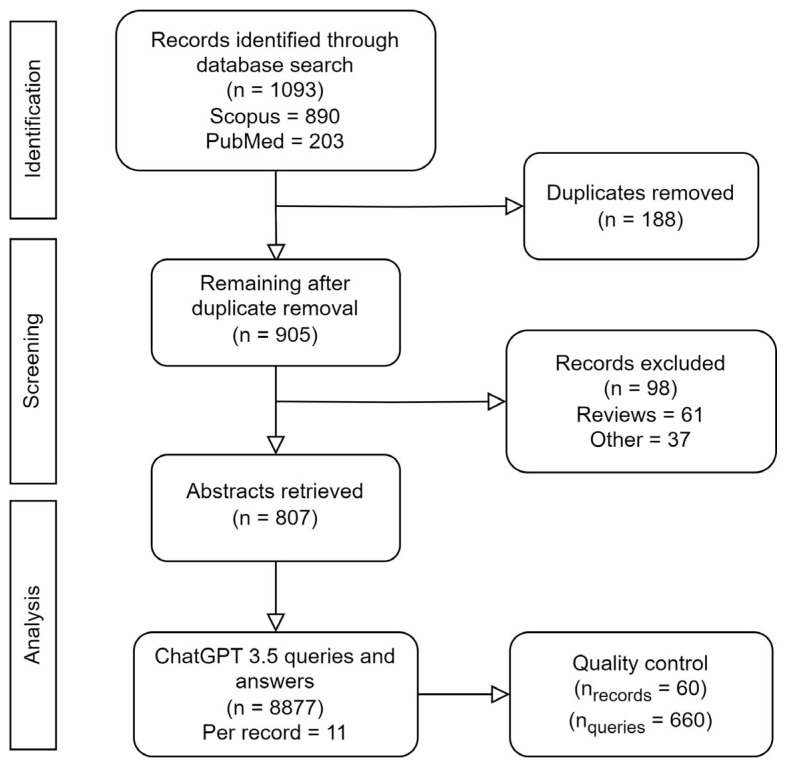
Flowchart of the paper selection and analysis process.

**Figure 2 cancers-15-04389-f002:**
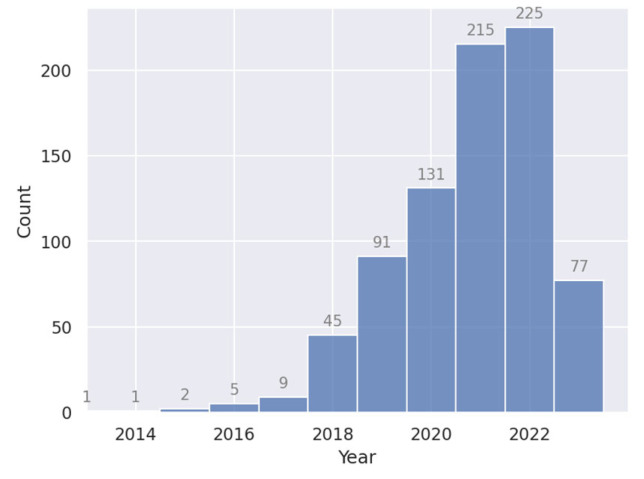
Number of deep learning medical image segmentation studies published per year. Not pictured: five papers published between 1999 and 2012.

**Figure 3 cancers-15-04389-f003:**
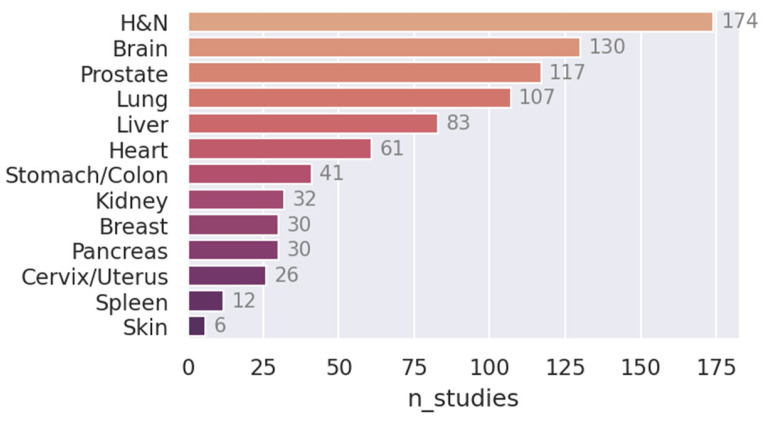
The total number of published studies per cancer site.

**Figure 4 cancers-15-04389-f004:**
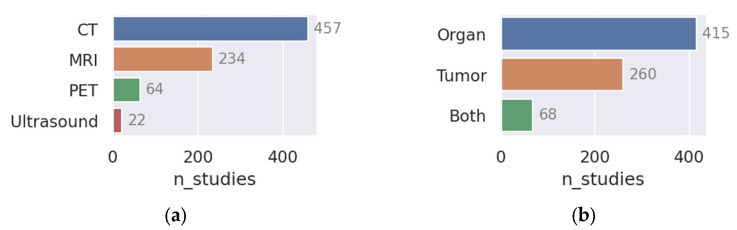
The total number of published studies organized by (**a**) image type and (**b**) segmentation type.

**Figure 5 cancers-15-04389-f005:**
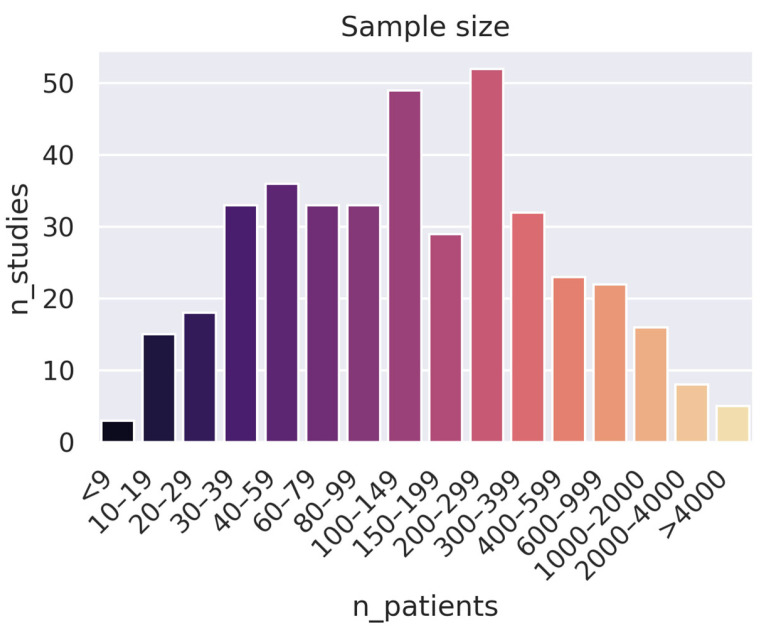
Distribution of the number of patients included in the studies (if a study used multiple datasets, their sizes were added together).

**Figure 6 cancers-15-04389-f006:**
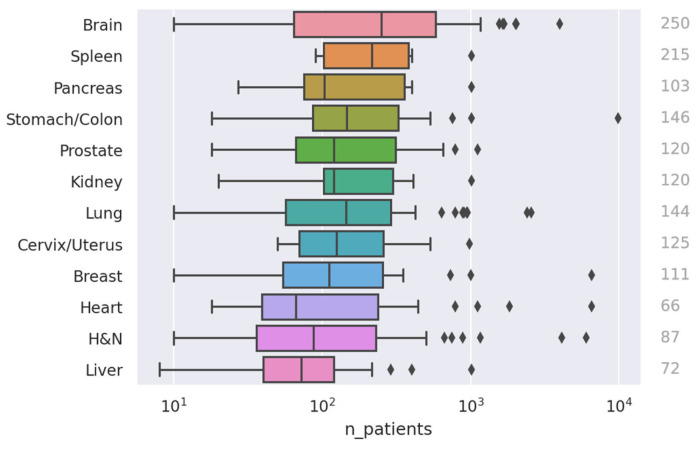
Distribution of dataset sizes per body region. The numbers to the right indicate the median sample size. Note the log scale on the *x*-axis.

**Figure 7 cancers-15-04389-f007:**
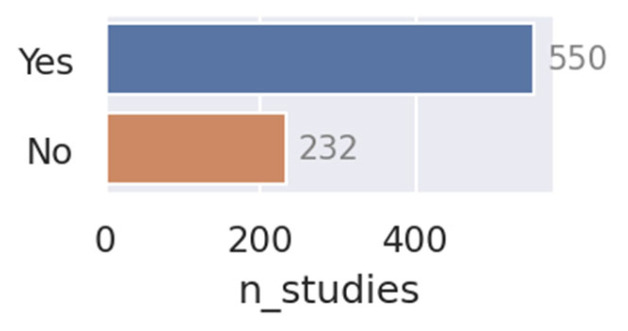
The number of yes and no responses when asked: “Did the paper propose a novel segmentation method or deep learning architecture? (Yes/No)”. The number of studies proposing novel models/architectures shows that the vast majority of papers focus on developing new methods.

## Data Availability

Data of the manuscript are available upon reasonable request.

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
