# Peer review of "Automatic Segmentation with Deep Learning in Radiotherapy"

_cancers, 2023, doi:10.3390/cancers15174389_

Round 1
Reviewer 1 Report
The paper's automatic segmentation with deep learning in radiotherapy needs a lot of improvements to enhance the paper's overall structure.
1) Add a related work section to your paper.
2) Clearly mention your contribution.
3) Summarize all the selected papers and make proper sections with tables of every tumor individually, only mentioning a number of papers relating to the specific tumor is not enough. I am mentioning the paper for your reference.
· https://doi.org/10.1016/j.engappai.2023.106276
4) Make a proper systematic review flowchart of the paper selection process.
5) Discussed the different image types (PET/CT/MRI) pros and cons, and which modality is often used in medical image segmentation and why? Moreover, discuss the advantages of using single modality vs multimodality, and point out where single and multimodality are used in your reviewed papers.
6) Include different datasets related to specific cancer, and mention whether it is public or private. Also, include the number of samples for training and testing in each dataset.
7) Discussed which parameters are useful in building a good segmentation model.
Author Response
Reviewer 1
The paper's automatic segmentation with deep learning in radiotherapy needs a lot of improvements to enhance the paper's overall structure.
- Add a related work section to your paper.
Response: Thank you for the comment, this has been added to Section 2.
2) Clearly mention your contribution.
Response: Thank you for the comment, our contribution has been added to Section 2.
3) Summarize all the selected papers and make proper sections with tables of every tumor individually, only mentioning a number of papers relating to the specific tumor is not enough. I am mentioning the paper for your reference.
- https://doi.org/10.1016/j.engappai.2023.106276
Response: Thank you for your comment. Our intention was never to scrutinize individual papers this deeply, which is why we did not include tables as the ones described here. That being said, the paper in the link is great and highly relevant to our work.
4) Make a proper systematic review flowchart of the paper selection process.
Response: Thank you for the suggestion, we have added the flowchart (Figure 1).
5) Discussed the different image types (PET/CT/MRI) pros and cons, and which modality is often used in medical image segmentation and why? Moreover, discuss the advantages of using single modality vs multimodality, and point out where single and multimodality are used in your reviewed papers.
Response: The appropriate imaging modality is typically determined by the clinical workflow leading up to the segmentation process. Often, there is only a single modality available, so comparing the modalities head-to-head is not generally possible or fruitful. In terms of multi-modality, the issue is rather nuanced, and we believe it is out of scope for what our work is trying to accomplish.
6) Include different datasets related to specific cancer, and mention whether it is public or private. Also, include the number of samples for training and testing in each dataset.
Response: This would be an interesting inclusion, but we decided to leave it as is for now, seeing as it would potentially be quite a big inclusion. Perhaps a dedicated review for different aspects of the data could be a future research project.
7) Discussed which parameters are useful in building a good segmentation model.
Response: This is well outside the scope of our work. The parameters vary greatly and should be carefully considered on a case-by-case basis.
Reviewer 2 Report
The manuscript presents a review of current automatic segmentation studies utilizing deep learning in the context of radiotherapy planning. The review encompasses 807 published papers, covering various cancer sites, image modalities (CT/MRI/PET), and segmentation techniques. The goal is to identify trends, commonalities, gaps, and areas for further research in this domain. The authors also employ explicit questions to analyze the corpus and provide practical insights. The review aims to guide researchers in conducting effective segmentation studies within the competitive landscape of medical image segmentation.
Congratulations on a well-structured and informative manuscript. The idea of using explicit questions for analysis adds a practical dimension to the review, making it a valuable resource for researchers seeking actionable insights. The use of ChatGPT for information condensation is a novel approach that highlights the manuscript's innovation. The provision of a supplementary spreadsheet with paper abstracts and summaries enhances the accessibility and usability of the review. The manuscript successfully addresses the challenge of keeping up with developments in automatic segmentation studies and offers valuable guidelines for conducting future research. Well done!
Author Response
Response: We thank the reviewer for taking their time to evaluate our work and provide feedback.
Reviewer 3 Report
The authors propose an interesting review regarding the use of deep learning-based automatic segmentation in radiotherapy. The English language is correctly and adequately used, the methodology is not defined as systematic (and coherently no PRISMA flow-chart and checklist are included) but well explained, the results are well illustrated and justify the conclusions and the discussion, complete with an evaluation of the study's limitations, in relation to the main reference literature.
My only suggestion to the authors: include in the main text a short paragraph regarding the limitations of the paper to confirm and make explicit that the review is not systematic.
Author Response
Response: Thank you for the comment. We have added a paragraph about limitations as suggested by the reviewer before the conclusion section. Note that, while the review is not strictly systematic, we found it useful to include a workflow diagram anyway, as suggested by reviewer #1.
Round 2
Reviewer 1 Report
Most of my comments are addressed. I recommend acceptance of this article in present form.